

# Nutrient availability induces community shifts in seagrass meadows grazed by turtles

Isis Gabriela Martínez López[1,2], Marloes van Den Akker[3], Liene Walk[3], Marieke M. van Katwijk[3], Tjisse van Der Heide[3] and Brigitta I. van Tussenbroek[1]

[1] Instituto de Ciencias del Mar y Limnología/Unidad Académica de Sistemas Arrecifales-Puerto Morelos, Universidad Nacional Autónoma de México, Puerto Morelos, Quintana Roo, Mexico
[2] Posgrado en Ciencias del Mar y Limnología, Universidad Nacional Autónoma de México, Delegación Coyoacán, Ciudad Universitaria, Ciudad de México, Mexico
[3] Institute for Water and Wetland Research, Radboud University, Nijmegen, The Netherlands

Corresponding author
Isis Gabriela Martínez López,
igml_2015@comunidad.unam.mx

## ABSTRACT

In the Caribbean, green turtles graze seagrass meadows dominated by *Thalassia testudinum* through rotational grazing, resulting in the creation of grazed and recovering (abandoned) patches surrounded by ungrazed seagrasses. We evaluated the seagrass community and its environment along a turtle grazing gradient; with the duration of (simulated) grazing as a proxy for the level of grazing pressure. The grazing levels consisted of Short-term (4 months clipping), Medium-term (8 months clipping), Long-term grazing (8 months of clipping in previously grazed areas), 8-months recovery of previously grazed patches, and ungrazed or unclipped patches as controls. We measured biomass and density of the seagrasses and rhizophytic algae, and changes in sediment parameters. Medium- and Long-term grazing promoted a shift in community species composition. At increasing grazing pressure, the total biomass of *T. testudinum* declined, whereas that of early-successional increased. Ammonium concentrations were highest in the patches of Medium-term ($9.2 + 0.8$ μM) and Long-term grazing levels ($11.0 + 2.2$ μM) and were lowest in the control areas ($4.6 + 1.5$ μM). *T. testudinum* is a late-successional species that maintains sediment nutrient concentrations at levels below the requirements of early-successional species when dominant. When the abundance of this species declines due to grazing, these resources become available, likely driving a shift in community composition toward a higher abundance of early-successional species.

## INTRODUCTION

Changes in the species composition of seagrass communities have been primarily attributed to bottom-up control mechanisms, such as resource availability (*Touchette & Burkholder, 2000*; *Ralph et al., 2007*), with top-down mechanisms, such as herbivory, only playing a minor role. The drastic declines of large herbivores, like green turtles, manatees, and dugongs, caused by human overexploitation (*Jackson, 1997, 2001*; *Hughes et al., 2004*; *Valentine & Duffy, 2006*) has contributed to the undervaluation of top-down

controls in seagrass communities. In recent decades, conservation strategies have led to local increases in abundance of green turtles (*Chelonia mydas*), with noticeable impacts of turtle grazing on seagrass communities (*Zieman, Iverson & Ogden, 1984*; *Kaladharan et al., 2013*; *Kelkar et al., 2013*; *Molina-Hernández & Van Tussenbroek, 2014*). Herbivores, particularly large herbivores such as green turtles, can alter plant species composition and community structure (*Lal et al., 2010*; *Kelkar et al., 2013*; *Molina-Hernández & Van Tussenbroek, 2014*).

Herbivore-induced shifts in community dynamics, structure and species composition have been widely studied for terrestrial ecosystems (*Bowes, 1993*; *Anderson & Briske, 1995*; *Augustine & McNaughton, 1998*). Large herbivores may alter plant communities through numerous mechanisms, such as selective grazing which alters competitive interactions among plant species (*Anderson & Briske, 1995*). The effects of grazing can be particularly strong when herbivores decrease the abundance of one or more dominant plant species that control resource abundance within the community (*Olff & Ritchie, 1998*). In grasslands for example, ungulates directly influence the nitrogen cycle by adding nitrogen through urine and feces, but more importantly they indirectly affect decomposition processes in soil through changes in plant litter. Resource heterogeneity created by ungulate grazing may alter the competition for resources, leading to an increase in plant species diversity in grazed areas (*Hobbs, 1996*; *McNaughton, Banyikwa & McNaughton, 1997*; *Bakker, Blair & Knapp, 2003*; *Borer et al., 2014*). Interactions of herbivores on plant communities can also be influenced by the effects of abiotic sources of disturbance, such as fire in terrestrial grasslands (*Hobbs, 1996*).

In the Caribbean, climax seagrass communities are dominated by the robust late-successional species *Thalassia testudinum* (slower-growing), which is the preferred food source for green turtles (*Bjorndal, 1980*; *Thayer et al., 1984*; *Molina-Hernández & Van Tussenbroek, 2014*). Green turtles feeding on *T. testudinum* use a cultivation or rotational feeding strategy (*Molina-Hernández & Van Tussenbroek, 2014*), which is also used by roaming wild ungulates in terrestrial systems (*Vavra & Ganskopp, 1998*; *Bakker et al., 2016*). Green turtles create grazing patches within *T. testudinum* meadows which are easily recognized as they graze seagrass leaves in specific areas or patches that they maintain and revisit (*Bjorndal, 1980*; *Molina-Hernández & Van Tussenbroek, 2014*). Continuous grazing on *T. testudinum* increases nitrogen content in the leaves and reduces their lignin content, which improves the quality of the food (*Bjorndal, 1980*; *Thayer et al., 1984*; *Zieman, Iverson & Ogden, 1984*). Rotational grazing by green turtles decreases the above-ground community biomass, particularly that of *T. testudinum*. Grazed patches are maintained for 1–2 years after they are abandoned most likely because of reduced leaf growth due to internal carbohydrate depletion (*Fourqurean et al., 2010*; *Lacey, Collado-Vides & Fourqurean, 2014*), and the turtles do not return to abandoned patches (*Molina-Hernández & Van Tussenbroek, 2014*). Abandoned patches show thinner and shorter leaves of *T. testudinum*, with sparser seagrass shoots than ungrazed nearby meadows (*Molina-Hernández & Van Tussenbroek, 2014*). Rotational grazing creates gaps (patches) allowing for colonization of faster-growing seagrass species such as *Halodule wrightii* and rhizophytic algae (early-successional species). Thus, in these gaps, turtle

herbivory drives community shifts with faster-growing species replacing slower-growing dominant species (*Molina-Hernández & Van Tussenbroek, 2014*, *Lacey, Collado-Vides & Fourqurean, 2014*). After abandonment, the recovery of grazed patches to pre-grazing conditions may take several years (*Molina-Hernández & Van Tussenbroek, 2014*).

Seagrass community structure and dynamics are highly influenced by competition for light (*Fourqurean et al., 1995*; *Ralph et al., 2007*) and nutrients (*Williams, 1987*; *Williams, 1990*; *Fourqurean et al., 1995*; *Touchette & Burkholder, 2000*; *Lee, Park & Kim, 2007*; *Leoni et al., 2008*). In the shallow seagrass beds in Caribbean reef lagoons, light availability is commonly high (*Enríquez & Pantoja-Reyes, 2005*) but when *T. testudinum* is dominant, the canopy itself reduces light availability under de canopy (*Williams, 1987*; *Enríquez & Pantoja-Reyes, 2005*). However, in grazed patches *T. testudinum* leaves are shorter and abandoned patches have less dense canopy and thinner *T. testudinum* leaves, and the reduced seagrass canopies in the grazed and abandoned patches are not expected to attenuate light, so competition for light is unlikely in turtle-grazed areas. Instead, nutrient availability in Caribbean seagrass meadows is fundamental in the process of succession described by *Williams (1990)*. *Williams (1990)* reported that during the first stages of succession, faster-growing seagrass species and rhizophytic algae are the primary successive species to recruit to new areas which stabilize sediments and increase organic matter content, and ammonium concentration in pore water. Slower-growing *T. testudinum* increases gradually in abundance, and when *T. testudinum* becomes dominant, ammonium concentration drops again because this climax species withdraws ammonium until levels below the requirements of early-successional species. Plant in the communities influence nutrient availability, but also changes in nutrient supply influence the species composition of the community. For example, when nutrient supply increases (eutrophication), an increase in the faster-growing seagrass species, such as *Halodule wrightii* or *Syringodium filiforme* and rhizophytic macroalgae may occur, eventually replacing *T. testudinum* (*Fourqurean et al., 1995*; *Davis & Fourqurean, 2001*; *Ferdie & Fourqurean, 2004*).

To date, there is little data on the underlying mechanisms associated with community shifts due to turtle grazing. *Fourqurean et al. (2010)*, *Ballorain (2010)* and *Christianen et al. (2011*, *2014)* studied changes in seagrass communities when recovering from grazing after placement of turtle exclusion cages; but the recovery trajectory not necessarily is the precise reverse of the trajectory of impact. Thus, following changes in the seagrass meadow when grazing pressure increases may be a better approach to discern the main drivers for the shifts in the seagrass vegetation. In the present study, we monitored changes in patches that were clipped/grazed for different duration (which we consider equivalent to grazing pressure). By comparing different levels of grazing pressure (grazing duration), we aim to answer two questions: (i) does rotational grazing change nutrient availability in sediments? and (ii) if such changes occur, what are the consequences for the species composition of the vegetation? We hypothesize that early-successional rhizophytic algae and seagrass species (faster-growing) will become more abundant if nutrient availability changes under increasing grazing pressure.

## MATERIALS AND METHODS

### Study site

The study was carried out in Puerto Morelos reef lagoon, on the northeastern coast of the Yucatan Peninsula, Mexican Caribbean. The reef lagoon is delimited by a fringing reef that extends parallel to the coast from north to south. The lagoon is governed by marine conditions, and is between 550 and 3,000 m wide, with a mean depth of three to four m (*Instituto Nacional de Ecología INE, 2000*; *Rodríguez-Martínez et al., 2010*). The calcareous sandy bottom is covered with a mixed macrophyte community dominated by *T. testudinum*, accompanied by *S. filiforme*, *Halodule wrightii* and rhizophytic algae (i.e., *Halimeda* spp. *Penicillus* spp., *Rhipocephalus* spp., *Udotea* spp.; *Van Tussenbroek, 2011*). While most of the extensive seagrass bed is not grazed by turtles, specific locations are frequently visited by them. In these locations, turtles create a mosaic of grazed (grazed continuously), ungrazed and recently abandoned patches (*Molina-Hernández & Van Tussenbroek, 2014*). The site of this study was such an area located 650 m from the coast (20°51′44.1″N, 86°51′46″W), at a depth of three to four m.

### Simulated turtle grazing

We established four levels of grazing pressure (based on grazing duration): Control as a reference condition, Short-term, Medium-term and Long-term grazing, and one level of Recovery (Table 1) with five replicates (patches) for each level. The experimental patches were created in two different previous conditions: ungrazed or previously grazed by green turtles. Control patches were established in ungrazed sections of the seagrass bed without clipping. Short-term and Medium-term experimental patches were created also in the ungrazed seagrass bed. We simulated turtle grazing by "clipping" leaves of seagrasses (*T. testudinum* and *S. filiforme)* and thalli of rhizophytic algae ~3 cm above sediment level with scissors (which is the average size of shoot leaves after being grazed by green turtles), removing the clipped portions of the seagrasses and algae to mimic ingestion by turtles. Although *S. filiforme* and calcareous green algae are not preferred food for the turtles, they do crop them incidentally. In addition, our objective was to show how grazing of all species, without preference, can still lead to changes in competitive hierarchy and therefore species composition. In this way, we mimicked turtle patches with irregular shape and 6–12 m² in size, which is the minimal patch size registered by *Molina-Hernández & Van Tussenbroek (2014)* in the same reef lagoon. Distances between adjacent patches were ≥1.5 m, which corresponded with the median distance of naturally created grazing patches by turtles in the study site. Seagrasses were re-clipped at ~15 days intervals which coincide with the time needed for *T. testudinum* blades to regrow ≥5 cm above the blade/sheath junction (*Molina-Hernández & Van Tussenbroek, 2014*).

The Short- and Medium-term patches were clipped for 4 and 8 months, respectively. The Long-term grazing areas were created in previously grazed patches that had been abandoned by turtles, which were easily identified, presenting sparse vegetation of *T. testudinum* with thinner and shorter leaves than the surrounding ungrazed seagrass bed (*Molina-Hernández & Van Tussenbroek, 2014*); the leaves in these patches did not have

**Table 1 Grazing levels of the experiment, with dates of start and duration of the treatments.** Recovery level was marked in June 2015 and then was undisturbed till the end of the experiment (8 months). The experiments were finished in February–March 2016.

| Previous seagrass condition | Experimental clipping | Explanation | Initiation of clipping |
|---|---|---|---|
| Ungrazed | None | Control | |
| Ungrazed | 4 months | Short-term | October 2015 |
| Ungrazed | 8 months | Medium-term | June 2015 |
| Previously grazed* | 8 months | Long-term** | June 2015 |
| Previously grazed* | None | Recovery | |

Notes:
* Previously grazed patches had just been abandoned by the turtles; *Molina-Hernández & Van Tussenbroek (2014)* established that abandoned patches were grazed by turtles ≥1 y.
** The Long-term grazing level does not occur naturally in patch rotational grazing because turtles do not return to abandoned patches.

evidence of turtle bites which are clearly visible when present. Within the previously grazed patches, experimental quadrats of one m² were established and seagrasses and algae were clipped as described above for 8 months. The Long-term level represents a seagrass scenario under heavy grazing pressure. The remaining areas in the same patches were assigned for the Recovery level without clipping. At four occasions in different patches (out of 18 times that clipping was applied), the turtles visited our experimentally clipped patches, which it was evident by short-cropped leaves; manual clipping was ceased when this occurred.

## Sediment and pore-water analysis

Sediment samples were collected to determine organic matter content at the end of our grazing treatments in February 2016. Three samples were taken randomly at each patch using five cm diameter tubes at a depth of 10 cm. The samples were pooled per patch, homogenized after removing plant material, and dried at 60 °C (~48 h) to be weighed afterward. Subsamples of 20 dry g were used to determine organic matter content through loss of dry weight after dissolution in hydrochloric acid (12.5%), which dissolves the carbonate sediments. The remaining sediment samples were used to granulometric analysis (Particle Analyzer CAMSIZER, Institute of Engineering, UNAM).

Pore-water samples were taken anaerobically at each patch at four different times starting 1 month after cessation of clipping, on 3 March, 5, 11 April and 6 May 2016. A rhizon pore water sampler (0.2 μm pore size, five cm depth; Eijkelkamp Agri-Search Equipment, Giesbeck, the Netherlands) connected to a 50 mL syringe was introduced in the sediment, and for 90–120 min allowing the syringe to slowly fill with pore-water. In the laboratory, sulfide concentrations were measured in 4 mL of each sample within 4 h of collection. Sulfide concentrations were measured in a mixture of 50% sample and 50% sulfide anti-oxidation buffer (*Lamers, Tomassen & Roelofs, 1998*) with a HI 4115 Silver/ Sulfide combination electrode (Hanna Instruments, Woonsocket, RI, USA), and a WTW multimeter (multi 340i/SET). The remaining pore-water of the samples was frozen (−20 °C) until further analysis. Concentrations of phosphate ($PO_4^{3-}$), nitrate ($NO_3^-$) and ammonium ($NH_4^+$) were measured on an Analytical order 1000640-1 3 System AutoAnalyser (Bran&Lubbe, 2 Seal Model II systems; continuous analysis; Norderstedt,

Germany) with ammonium molybdate, hydrazine sulfate and salicylate, respectively. Detection limits for the analysis was one μM for sulfide concentrations, 0.2 μM, for ammonium concentrations, and 0.1 μM for nitrate and phosphate concentrations.

## Seagrass and rhizophytic algae

Samples were obtained with permit PPF/DGOPA-012/17 of Secretaría de Agricultura, Ganadería, Desarrollo Rural, Pesca y Alimentación, México. At the end of the experiment (February–March 2016), foliar shoots of seagrasses and thalli of rhizophytic algae were counted for each species in a randomly placed PVC ring (diameter = 30 cm area = 0.07 m$^2$). In each patch, two core samples (diameter = 11.2 cm, $h$ = ~40 cm) were taken after 25 days of the last clipping, allowing for regrowth of leaves and thalli before harvesting. In patches with low density of *T. testudinum*, additional foliar shoots of *T. testudinum* were collected randomly by cutting the vertical rhizome below the substratum to complete 15 shoots for morphometric measurements. One of the core samples was processed for *T. testudinum* and the other for the early-successional plants (rhizophytic algae and *S. filiforme*).

In the laboratory, plant samples were stored in a refrigerator and processed within 15 days after collection. The core samples were rinsed and for each seagrass species, material was separated into below-ground roots, horizontal rhizome, vertical rhizome, sheaths and above-ground leaf sections. The leaves were cleaned of epiphytes with a razor blade. Complete thalli of rhizophytic algae were also selected from core samples. Thalli were rinsed but attached sand grains to the rhizoids were not removed. Epiphytes were also removed with a razor blade. The plant fractions and thalli of rhizophytic algae were placed in a drying oven for at least 36 h at 60 °C until completely dry, after which their dry weight was determined on an analytical balance. Dried horizontal rhizomes and leaves for *T. testudinum* were preserved for further analysis.

For each *T. testudinum* shoot, the width of the second youngest leaf was measured at the base with a dial caliper (precision 0.02 mm). Total horizontal rhizome length of *T. testudinum* per core sample was measured with a ruler.

## Nutrients and soluble carbohydrates analyses in late-successional seagrass species *T. testudinum*

Tissue nitrogen (N) and carbon (C) were measured from three mg dried and ground leaves of *T. testudinum* with a carbon-nitrogen-sulfur analyzer (type NA1500; Carlo Erba, Thermo Fisher Scientific, Waltham, MA, USA), coupled via an interface (Finnigan Conflo III) to a mass spectrometer (Thermo Finnigan Delta Plus, Waltham, MA, USA). Total concentration of phosphorous (P) in *T. testudinum* leaves, was measured with inductively-coupled- plasma emission spectrometry (IRIS Intrepid II; Thermo Electron Corporation, Franklin, MA, USA), after digestion with nitric acid, following *Smolders et al. (2006)*. Total soluble carbohydrates were measured in *T. testudinum* horizontal rhizomes. The sugars were extracted from ground rhizomes in ethanol at 80 °C. Extracts were then evaporated until dry at room temperature, under a stream of compressed air and subsequently re-dissolved in dH$_2$O. Samples were analyzed with a spectrophotometer using a hydrochloric acid/resorcinol assay standardized to sucrose (*Huber & Israel, 1982*).

## Statistical analysis

A one-way ANOVA was applied to analyze the effects of the different levels of grazing pressure on the percentage of organic matter content in sediments, and in the percentage of fine sediments (0.3–0.189 mm of grain size). Nested ANOVA analyses were used to test grazing effect on phosphate ($PO_4^{3-}$), nitrate ($NO_3^-$) and ammonium ($NH_4^+$) concentrations in pore-water samples, with pore-water samples per patch were nested within grazing level. Differences among levels were determined with a post hoc Tukey HSD tests. To test for grazing effects on seagrass community, a two-way ANOVA was applied to the total biomass and density per species group per patch, with grazing pressure and species (group) as fixed factors. *Syringodium filiforme* and rhizophytic algae were considered together as early-successional plants and *T. testudinum* as the late-successional species. Using a one-way ANOVA analysis, we also tested the relative contribution of early-successional species to the total density of the vegetation ((*S. filiforme* + rhizophytic algae)/(*S. filiforme* + rhizophytic algae + *T. testudinum*)). To test for grazing effects on *T. testudinum*, a one-way ANOVA was applied to the percentage of nitrogen, carbon, and phosphorous concentration in leaves, and for the soluble carbohydrates in rhizomes ($\mu$mol g DW$^{-1}$). Differences among levels were determined with a post hoc Tukey HSD tests. Density data were log transformed prior to analysis (log+1), and percentages were arcsine transformed prior to analysis. All analyses were conducted using Systat 11 software. Assumptions of normality and homoscedasticity were tested through graphical analysis of the residuals. For the two-way ANOVA analysis, two outliers in *T. testudinum* total biomass data and one for early-successional species were removed after residual analysis. In the same way, four outliers in density data of the early-successional species were removed. For the one-way ANOVA analysis of the relative contribution of early-successional species to density, one outlier was removed. To test the relationship of total biomass of *T. testudinum* and pore-water ammonium concentrations, a linear regression was performed using sub-replicates of ammonium measurements within each patch and the total biomass of this species in each patch. Five outliers were removed.

## RESULTS

### Simulated turtle grazing

We simulated green turtle rotational grazing as closely as possible at appropriate temporal and spatial scales. Re-clipping intervals mimicked natural turtle grazing rates and size of the experimental patches corresponded with the minimal size created by green turtles in the area. The fact that turtles grazed on four separate occasions in our clipped patches (which were different Medium-term patches) could be indicative that the manual manipulation to copy the grazing behavior by turtles was sufficiently realistic. Visits to our experimental patches were frequent and therefore we could always establish that turtles had visited the site; their grazing and our clipping were applied with almost the same frequency, thus creating a minimal bias.

### Sediment and pore-water analyses in grazing levels

Sediment analysis showed no differences in organic matter content among grazing levels ($F_{(4,20)} = 0.2523$ $p = 0.9049$; Table 2) while percentage of fine sediments were significantly

**Table 2 Average values (+SE) and results of sediment analysis.** One-way ANOVAs tested differences in the percentage of organic matter content and fine sediments (%) among grazing levels: No grazing (Control), Short-term, Medium-term, Long-term and Recovery. Data were arcsine transformed. Different letters represent significant pairwise comparisons among the levels.

| | Control | Short-term | Medium-term | Long-term | Recovery | ANOVA Results | | |
| | Mean ± SE | Mean ± SE | Mean ± SE | Mean ± SE | Mean ± SE | df | F | p |
|---|---|---|---|---|---|---|---|---|
| Organic matter (%) | 1.70 ± 0.32 | 1.56 ± 0.34 | 1.87 ± 0.33 | 1.52 ± 0.34 | 1.55 ± 0.53 | 4 | 0.252 | 0.905 |
| Finese diments (%) | 54.75 ± 3.14[a] | 53.67 ± 3.91[a] | 41.5 ± 11.03[a,c] | 7.83 ± 0.05[b] | 17.90 ± 6.27[c] | 4 | 11.299 | **<0.001** |

Note:
Significantly different at alpha = 0.05 (in bold type).

**Table 3 Nested ANOVAs used to test differences in ammonium $[NH_4^+]$, phosphate $[PO_4^{3-}]$ and nitrate $[NO_3^-]$ concentrations (μM) from sediment pore-water as a function of the grazing levels.** Grazing levels are Control ($n = 18$), Short-term ($n = 20$), Medium-term ($n = 20$), Long-term ($n = 19$) and Recovery ($n = 19$).

| Pore-water parameter | Nested ANOVA results | | | | |
| | SS | df | MS | F | p |
|---|---|---|---|---|---|
| Ammonium (μM) | | | | | |
| Treatment | 315.375 | 4 | 87.844 | 3.398 | **<0.013** |
| Subreplicate | 1,691.851 | 20 | 84.593 | 3.273 | **<0.001** |
| Residuals | 1,886.948 | 73 | 25.849 | | |
| Nitrate (μM) | | | | | |
| Treatment | 0.024 | 4 | 0.006 | 0.264 | 0.900 |
| Subreplicate | 0.294 | 20 | 0.015 | 0.648 | 0.862 |
| Residuals | 1.657 | 73 | 0.023 | | |
| Phosphate (μM) | | | | | |
| Treatment | 0.119 | 4 | 0.030 | 0.498 | 0.738 |
| Subreplicate | 1.558 | 20 | 0.078 | 0.207 | 0.207 |
| Residuals | 4.311 | 72 | 0.060 | | |

Note:
Significantly different at alpha = 0.05 (in bold type).

lower for the Long-term and Recovery levels ($F_{(4,20)} = 11.2993$ $p < 0.01$; Table 2). Ammonium concentrations in sediment pore-water were significant different among levels ($F_{(4,20)} = 3.398$ $p < 0.05$; Table 3) Ammonium concentrations ($NH_4^+$) were lowest for the Controls (mean ± SE = 4.6 ± 1.5 μM), and highest for the Medium- and Long-term levels (mean ± SE = 9.2 ± 0.8 μM and 11.0 ± 2.2 μM, respectively; Fig. 1). Pore-water nitrate ($NO_3^-$) and phosphate ($PO_4^{3-}$) concentrations were almost below detection level and no differences among levels were found (Table 3). Sulfide concentrations varied too much within and among levels with values below the detection limit (one μM) and a highest concentration of 382.5 μM; no statistical analyses were applied to these data.

## Seagrasses and rhizophytic algae response to grazing pressure

Responses to grazing levels of the macrophyte community were species-specific. The total biomass of *T. testudinum* decreased as the grazing pressure increased; it was highest

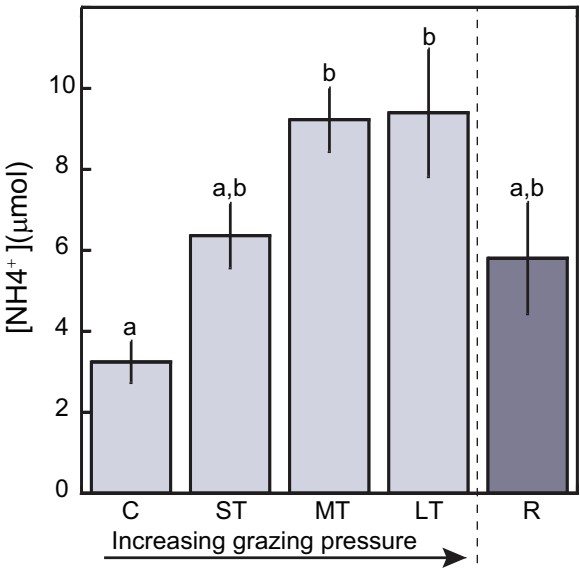

**Figure 1 Bar graphs displaying mean (±SE) ammonium concentration in sediment pore-water as function of grazing levels.** The grazing levels significantly affected ammonium concentration in sediment ($p = 0.013$), with different letters representing significant pairwise comparisons between treatments. C, Control; ST, Short-term; MT, Medium-term; LT, Long-term; R, Recovery.

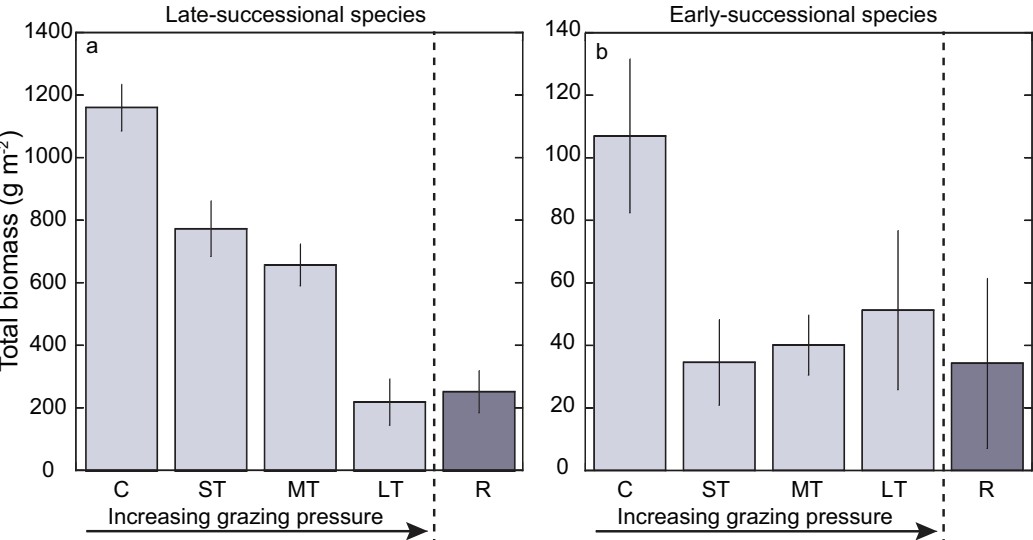

**Figure 2 Bar graphs displaying mean (±SE) total biomass (above- and below-ground).** (A) Late-successional species *Thalassia testudinum* and (B) early-successional species (*Syringodium filiforme* and rhizophytic algae), as a function of grazing levels. C, Control; ST, Short-term; MT, Medium-term; LT, Long-term; R, Recovery. Note differences in *Y*-axis

for the Controls (mean ± SE = 1,161.5 ± 73.6 gr DW m$^{-2}$), and lowest in the Long-term patches (mean ± SE = 220.1 ± 73.4 gr DWm$^{-2}$). *T. testudinum* biomass was slightly higher after 8 months of recovery from grazing but still lower when compared with

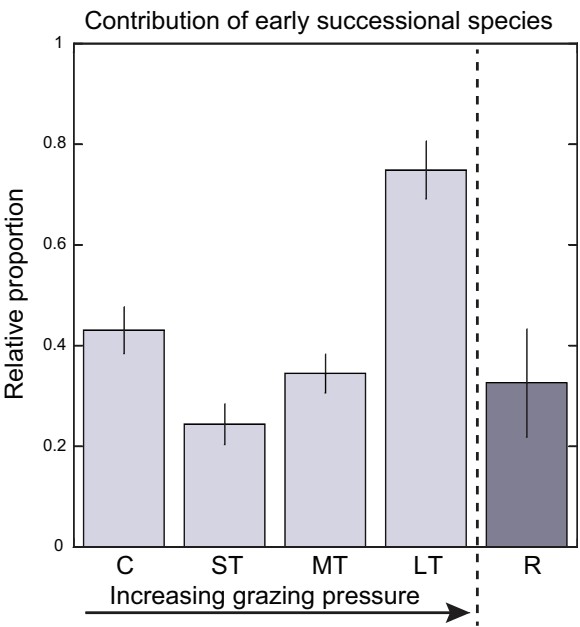

**Figure 3** **Bar graphs displaying mean (±SE) of the relative contribution of early-successional species (*Syringodium filiforme* and rhizophytic algae) to total density as function of grazing levels.** Grazing levels are C, Control; ST, Short-term; MT, Medium-term; LT, Long-term; R, Recovery.

Control level (Fig. 2). Trends with grazing pressure in the total biomass of the early-successional species (*S. filiforme* and rhizophytic algae) differed from those registered for the late-successional species, *T. testudinum*. Total biomass of the early-successional plants was higher for the Medium- and Long-term levels (mean ± SE = 40.1 ± 9.5 gr DW m$^{-2}$ and 51.3 ± 25.4 gr DW m$^{-2}$, respectively) than for the Short-term level (mean ± SE = 34.6 ± 13.6 gr DW m$^{-2}$), which had a decline in biomass of early-successional species in comparison with the Controls (mean ± SE = 107.0 ± 24.5 gr DW m$^{-2}$). Recovery level had similar values as the Short-term level (mean ± SE = 34.3 ± 27.1 gr DW m$^{-2}$) (Fig. 2). Trends in density were like those of biomass (Table S1). The two-way ANOVA analysis, testing for differences in total biomass per species group (early- and late-successional) and grazing levels resulted in a significant interaction between species group and levels ($F_{(4,37)}$ = 21.2575 $p$ < 0.01) confirming that the trends of changes in biomass differed between the late- and early-successional species for the grazing levels. The two-way ANOVA for density showed the same results ($F_{(4,36)}$ = 8.356 $p$ < 0.01). When we considered the relative contribution of early-successional species to the total density of vegetation we found that the contribution of *S. filiforme* and rhizophytic algae to the total density was significantly higher in the Long-term patches in comparison with the other grazing levels (one-way ANOVA ($F_{(4,19)}$ = 8.2385 $p$ < 0.01) Fig. 3; Table S2). The linear regression to test the relationship between ammonium availability and *T. testudinum* total biomass showed a significant inverse relationship ($r^2$ = 0.753 $p$ < 0.001 $n$ = 20). Ammonium concentrations in sediment increased in relation with a decrease in *T. testudinum* total biomass (Fig. 4).

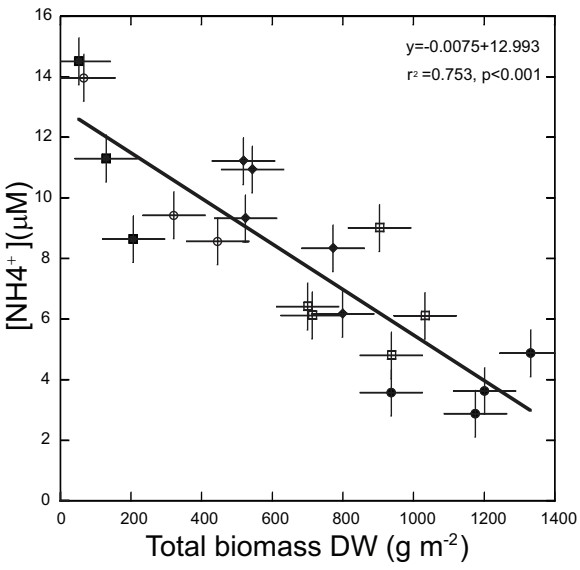

**Figure 4 Relationship between ammonium concentration in pore-water of sediments and total biomass of *Thalassia testudinum*.** Control (*filled circles*), Short-term (*open squares*), Medium-term (*filled diamonds*), Long-term (*filled squares*) and Recovery (*open circles*). Error bars ±SE.

**Table 4 Average values (+SE) and one-way ANOVA results used to test differences in *Thalassia testudinum* nitrogen, carbon and phosphorous content in leaves (%) and soluble carbohydrate reserves in rhizomes among grazing levels.**

| | Sample | Control | Short-term | Medium-term | Long-term | Recovery | ANOVA results | | |
|---|---|---|---|---|---|---|---|---|---|
| | | Mean + SE | Mean ± SE | Mean ± SE | Mean ± SE | Mean ± SE | df | F | p |
| Leaf nitrogen (N) | Shoot | 1.65 ± 0.04[a] | 2.02 ± 0.06[b] | 2.00 ± 0.04[b] | 2.00 ± 0.04[b] | 1.94 ± 0.04[b] | 4 | 7.394 | **<0.001** |
| Leaf carbon (C) | Shoot | 34.34 ± 0.73 | 34.8 ± 0.69 | 35.69 ± 0.83 | 35.3 ± 0.73 | 36.13 ± 0.47 | 4 | 1.180 | 0.351 |
| LeafC:N ratio | Shoot | 20.92 ± 0.81[a] | 17.3 ± 0.43[b] | 16.84 ± 0.86[b] | 17.63 ± 0.39[b] | 18.66 ± 0.57[a,b] | 4 | 5.511 | **<0.004** |
| LeafN:P ratio | Shoot | 18.27 ± 0.79 | 18.79 ± 1.00 | 20.87 ± 1.08 | 19.84 ± 1.35 | 20.03 ± 0.93 | 4 | 1.059 | 0.404 |
| Rhizome carbohydrates ($\mu M$ g $DW^{-1}$) | Core | 89.83 ± 14.41[a] | 60.6 ± 6.86[b] | 33.65 ± 2.66[c] | 9.79 ± 1.72[d] | 30.26 ± 4.53[e] | 4 | 77.020 | **<0.001** |

**Note:**
Significantly different at alpha = 0.05 (in bold type).

## Seagrass nutrient and carbohydrate content

Simulated grazing resulted in an increase of nitrogen (N%) in *T. testudinum* leaves. All grazing levels, including the Long-term level showed significant higher *N* concentrations when compared with control level ($F_{(4,17)} = 77.020$, $p < 0.01$). The *N* concentrations in the Recovery level also was significant higher compared to Control level. In contrast, Carbon (C%) did not show significant differences among levels ($F_{(4,19)} = 1.180$, $p = 0.35$). Consequently, the Control and Recovery level showed significant higher tissue C:N in *T. testudinum* when compared to the grazed levels (($F_{(4,19)} = 5.511$ $p < 0.01$) in C:N ratio). Contrary, N:P ratio did not show significant differences among grazing levels ($F_{(4,19)} = 1.059$, $p = 0.40$). Soluble carbohydrates content in *T. testudinum* horizontal rhizome had significant differences among all levels ($F_{(4,19)} = 5.511$, $p < 0.01$); it decreased at higher grazing pressure with a drastic decline in Long-term patches (Table 4).

## DISCUSSION

Large herbivores such as green turtles may be important agents of change in ecosystems. Rotational grazing by turtles has negative effects on the above-ground community biomass and causes changes in species composition of the vegetation. We found changes in the species composition and pore-water ammonium content with increasing grazing pressure, suggesting that the continuous removal of above-ground biomass of the dominant *T. testudinum* enhances nutrient availability, which in turn drives the shift in the species composition of the vegetation (Fig. 5).

### Changes in *Thalassia testudinum* under a rotational grazing regime

Simulated turtle grazing increased food quality, as nitrogen (N) content in *T. testudinum* leaves increased, which was also reported by *Bjorndal (1980)*, *Moran & Bjorndal (2007)* and *Molina-Hernández & Van Tussenbroek (2014)*. Phosphorus (P) content instead did not change among grazing levels. *Holzer & McGlathery (2016)* studied responses of *T. testudinum* to grazing in a phosphorus limited environment in Bermuda in plots with and without artificial fertilizers; and found that the cultivation grazing response of the turtles depended on the availability of phosphorus. In carbonate sediments phosphorus (P) is often the growth-limiting nutrient of many tropical seagrasses (*Fourqurean et al., 1995*). For seagrasses, tissue N:P ratio reflects the relative availability of these elements in the environment. In *T. testudinum* leaves, a N:P ratio around 30:1 is present when there is a critical level and balance in the availability of both nutrients (*Fourqurean & Zieman, 2002*). As the N:P ratios of this study were never higher than 21, we can assume that P was not limiting in our experiments. *Molina-Hernández & Van Tussenbroek (2014)* found much higher N:P ratios (>34) in turtle-grazed areas within the same reef lagoon in 2011. Differences between the two study areas in terms of P availability within the lagoon maybe due to differences in location or may be attributed to the massive influx of *Sargassum* spp. in 2015, during which enormous amounts of organic matter and nutrients were imported, resulting in higher P availability, and therefore higher P concentrations in seagrass tissues throughout the lagoon (*Van Tussenbroek et al., 2017*). Thus, N and not P was limiting to *T. testudinum* (and likely the other seagrasses), during our study.

The leaf N content was similar at all levels of grazing. The Long-term patches were previously grazed patches abandoned by the turtles. Thus, depletion of nutrient (N) content in leaves was not the reason for turtles to abandon the patches. The paradigm about N depletion in tissues as a patch abandonment cue is not supported by our data nor to those reported in previous studies (*Moran & Bjorndal, 2007*; *Fourqurean et al., 2010*). Turtles most likely stopped grazing because not much new tissue was formed, as shoot density decreased (this study) or the leaves stopped growing as suggested by *Lacey, Collado-Vides & Fourqurean (2014)*, which was likely driven by the decline in soluble carbohydrates content (Table 4, *Moran & Bjorndal, 2007*). Robust seagrass species like *T. testudinum* have a large proportion of tissue in non-photosynthetic plant parts such as roots and rhizomes that are maintained by photosynthesis in the leaves, which are reduced due to grazing (*Williams, 1988*; *Molina-Hernández & Van Tussenbroek, 2014*; *Bakker et al., 2016*). At first, the carbohydrate reserves in the rhizomes are used to maintain the

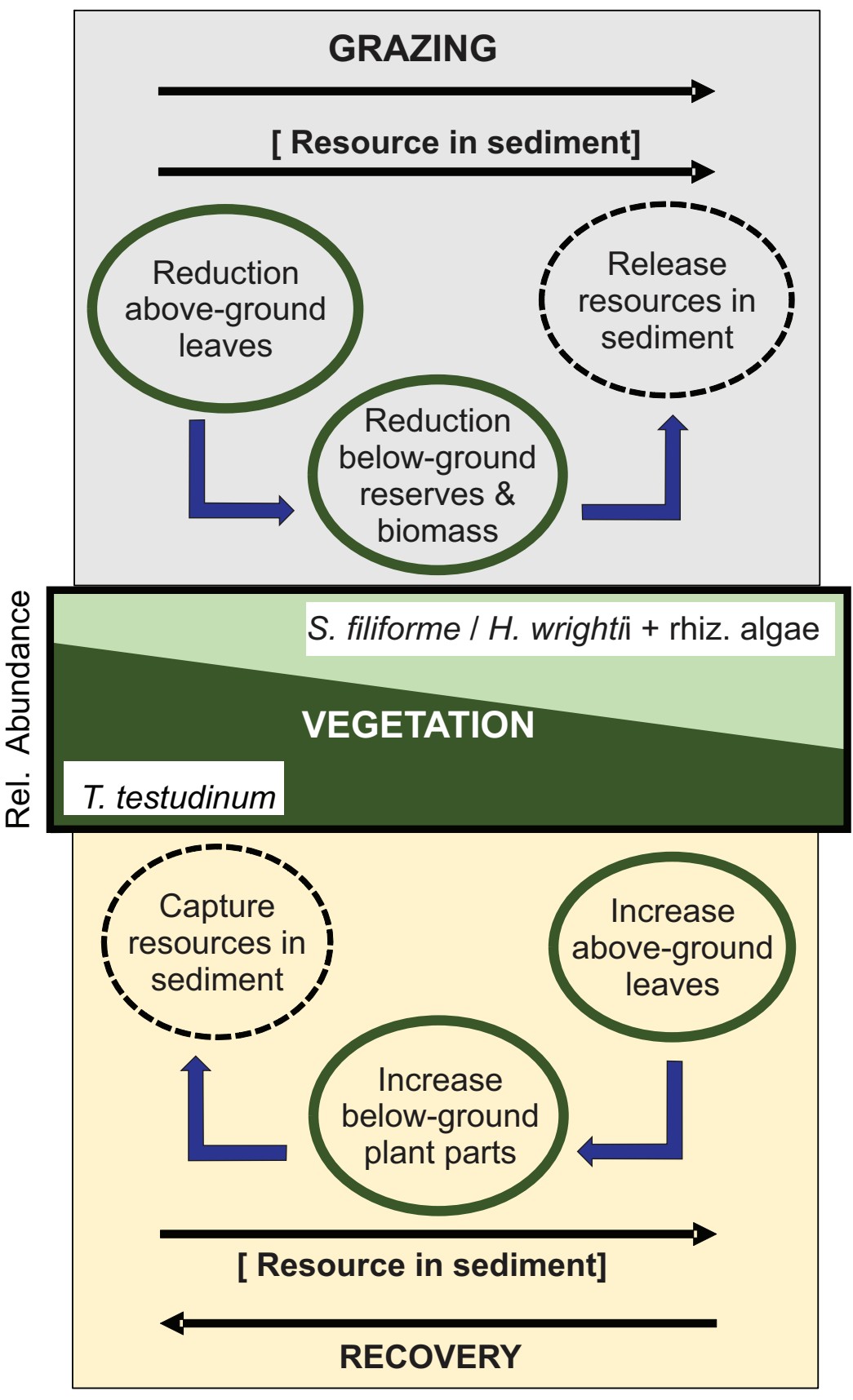
**Figure 5 Shift in species composition of a Caribbean seagrass community under a regime of rotational grazing, showing the principal processes involved in the transition from late seral state to earlier seral state during grazing and vice versa during recovery.** Resource in the sediment may be N, P, Fe, depending which is limiting under the prevailing conditions. Tt, *Thalassia testudinum*; Sf, *Syringodium filiforme*; Hw, *Halodule wrightii*; rhiz. algae rhizophytic algae.

metabolism of below-ground structure, but when these are depleted, below-ground tissues are lost resulting in a decrease in shoot density. Other indicators of the stress of grazing on seagrasses are reduced leaf length (direct effect of grazing) and width (most likely due to metabolic stress) as reported by *Moran & Bjorndal (2005)*, *Fourqurean et al. (2010)* and *Molina-Hernández & Van Tussenbroek (2014)*. The widths of the leaves increased slightly in the Recovery level, but after 8 months of recovery the leaves were still narrower than those of the Control level. These data emphasize the detrimental effect of prolonged turtle grazing on the shoots of *T. testudinum*.

## Does rotational grazing change nutrient availability in sediments?

In terrestrial and marine systems, the impact of grazing on plants results from both top-down (grazing) and bottom-up effects (nutrients; *De Mazancourt, Loreau & Abbadie, 1998*; *Holzer & McGlathery, 2016*). Ammonium concentrations in pore-water sediments showed the inverse tendency to *T. testudinum* density and biomass; which decreased when grazing pressure increased. *T. testudinum* is a late-successional species that as succession progresses outcompetes early-successional species by withdrawing nutrients below the earlier seral species requirement levels (*Williams, 1990*). The decrease in *T. testudinum* biomass induced by grazing pressure increased ammonium availability in pore-water, likely because less ammonium was utilized by *T. testudinum*. This was supported by the inverse relationship between pore-water ammonium concentrations and *T. testudinum* total biomass.

In terrestrial systems, nutrient input from urine and feces may also alter nutrient cycling (*Hobbs, 1996*; *Bakker, Blair & Knapp, 2003*). However, in marine systems green turtles often defecate at their resting areas when they become active (*Bjorndal, 1980*), and turtle feces floats, so urine and feces may not make a substantial contribution to changes in ammonium availability within sediment of grazed patches (*Thayer et al., 1984*; *Moran & Bjorndal, 2005*; *Moran & Bjorndal, 2007*). Changes of organic matter in sediments are also expected to influence nutrient availability. We expected that simulated grazing, especially in Long-term grazed patches, would result in decreased organic content of sediments, as turtles removed leaf material, and the short-cut grazed leaves were less likely to attenuate waves which increases trapped organic matter (*Thayer et al., 1984*; *Moran & Bjorndal, 2005*; *Christianen et al., 2011*). However, we did not find differences in organic sediment content among the grazing levels, even though the capacity to trap fine particles was reduced in grazed levels, as indicated by the lower proportion of fine sediments in the Long-term grazed and Recovery patches. Most likely, dying below-ground tissues build-up at increased grazing levels, contributing to organic matter in the sediments. Also, *Moran &*

*Bjorndal (2007)* found no changes in organic matter content in sediments after 16 months of simulated turtle grazing, with the effects of grazing on sediments possibly being site and species-dependent.

## Can changes in the macrophyte community composition by selective turtle grazing be associated with nutrient availability?

Rotational grazing reduced the total biomass of the macrophyte community, but the trends of change differed for early- and late-successional species, resulting in a shift in community toward faster-growing early-successional species, also reported by *Molina-Hernández & Van Tussenbroek (2014)*. While *T. testudinum* decreased with prolonging grazing pressure, early-successional species decreased in abundance at Short-term grazing pressure, remained stable during Medium-term grazing, but increased when grazing was Long-term. *S. filiforme* is an early-successional species crop occasionally and incidentall**y** by green turtles. Our Short-term grazing level showed a decrease in *S. filiforme* biomass, probably due to its thinner rhizomes and less carbohydrate reserves to compensate the losses of photosynthetic tissue (*Molina-Hernández & Van Tussenbroek, 2014*), but they can also occupy open spaces and utilize available resources faster than *T. testudinum* because they have higher rhizome elongation rates (*Williams, 1987*; *Marbà & Duarte, 1998*). Grazing has been shown to disrupt competitive hierarchies (*Anderson & Briske, 1995*), by compromising a species ability to cope with its competitors or by facilitating growth potentials of otherwise subdominant plant species. Changes in seagrass dominance in relation with mega herbivore grazing in tropical seagrasses have been reported before (*Lal et al., 2010*; *Lefebvre et al., 2017*). *Lefebvre et al. (2017)* reported that exclusion cages established in a manatee grazing area led a mixed seagrass community of *S. filiforme* and *H. wrightii* (both important in manatee diets) to shift to a dominant *S. filiforme* seagrass bed. They suggested higher rates of production, lateral branching and space occupation improved the competitive superiority of *S. filiforme* over that of *H. wrightii* for acquisition of nutrients, occupation of space and anchorage in sediments. Our study showed that when *T. testudinum* declined at higher grazing pressure, ammonia concentrations in sediments increased. Therefore, this nutrient increase likely contributed to the competitive balance in favor to *S. filiforme* and rhizophytic algae. This process is reversed during recovery (Fig. 5).

## Implications

Large herbivores like green turtles can have significant impacts as ecosystem engineers. Rotational grazing by turtles alters plant productivity and habitat structure (*Moran & Bjorndal, 2005*; *Molina-Hernández & Van Tussenbroek, 2014*). However, on a landscape-scale, it increases species and structural diversity (*Molina-Hernández & Van Tussenbroek, 2014*), by creating gaps allowing for the colonization of early-successional species like *S. filiforme*, *H. wrightii* and rhizophytic algae. Changes in morphology and species richness of seagrasses have been related with changes in the abundance and diversity of their associated fauna (*Ray et al., 2014*); and whether this applies to meadows under a rotational-grazing regime merits further investigation.

When turtles reach a density beyond the carrying capacity for turtle grass recovery (represented by our Long-term grazing level), they drive species replacement of a whole seagrass bed toward a higher dominance of faster-growing and early successional seagrasses (*Molina-Hernández & Van Tussenbroek, 2014*), inducing a pattern shift through meadows in lower successional stages (*Kelkar et al., 2013*) or cause complete loss of the meadow (*Fourqurean et al., 2010*). Herbivorous sea turtles may act as switches controlling transitions between alternative ecosystem states as they can affect the susceptibility of the ecosystem to abiotic disturbances, like ungulates in terrestrial systems. Ungulates may provide a switch between a fire-prone or a fire-resistant ecosystem, which also depends on environmental factors controlling primary productivity (*Hobbs, 1996*). Turtle grazing is also increasing as a consequence of successful conservation of green turtles and the absence of top predators like sharks (*Fourqurean et al., 2010*; *Christianen et al., 2014*; *Heithaus et al., 2014*; *Molina-Hernández & Van Tussenbroek, 2014*). These trends could significantly impact seagrass meadows, and consequences of reduction or loss of *T. testudinum* maybe synergistic with other disturbances (*Van Tussenbroek et al., 2014*). For example, *T. testudinum* is a deeply rooted species with a well-developed below-ground rhizome-root system and resists hurricanes better than faster-growing seagrass species with less below-ground biomass (*Cruz-Palacios & Van Tussenbroek, 2005*). Reduction of *T. testudinum* by grazing may thus enhance the vulnerability of the total seagrass bed to hurricanes.

Our study shows that turtle grazing influences nutrient cycling, likely by reducing the abundance of the dominant competitor for nutrients in the sediments, thereby allowing for an increase in abundance of faster-growing species that take advantage of the newly available resources (Fig. 5). Eutrophication, now widespread in the Caribbean, has similar consequences for seagrass plant communities (*Van Tussenbroek et al., 2014*). Turtle grazing herbivory may potentially threaten stability of meadows throughout the Caribbean; especially in synergy with other human-induced or natural stressors like eutrophication and hurricanes. *Christianen et al. (2018)* provided evidence of green turtle grazing as a main factor contributing to invasion of the non-native seagrass species *Halophila stipulacea* in the Caribbean.

## CONCLUSIONS

Rotational grazing by turtles has negative effects on the above-ground community biomass and causes changes in species composition of the vegetation. Our study showed that when *T. testudinum* biomass declined at higher grazing pressure, ammonium concentrations in the pore-water sediment increased, suggesting that the continuous removal of above-ground biomass of the dominant *T. testudinum* enhances nutrient availability in sediment. We found that even simulated grazing of all species (*S. filiforme* and rhizophytic algae) without preference, can still lead to changes in competitive hierarchy and therefore species composition. Therefore, green turtles may be important agents of change in ecosystems. When turtles reach a density beyond the carrying capacity for turtle grass recovery, they may drive species replacement of a whole seagrass bed inducing a pattern shift through meadows in lower successional stages or may cause the complete loss of a meadow. In this

way, herbivorous sea turtles act as switches controlling transitions between alternative ecosystem states as they can affect the susceptibility of the ecosystem to abiotic disturbances, like ungulates in terrestrial systems. To establish better conservation strategies, it is necessary to continue studying the interactions between turtle grazing and seagrasses, including the implications of seagrass community shifts due to turtle grazing to the stability and maintenance of ecosystem services of the seagrass meadows.

## ACKNOWLEDGEMENTS

Guadalupe Barba Santos, Hazel M. Canizales Flores, Luuk Leemans, Nancy E. Burgos Veneroso supported this study in the field. The Academic Service of Meteorological and Oceanographic Monitoring (SAMMO) provided meteorological data. Useful comments and input during development of this manuscript from Darren Brown are acknowledged.

### Funding

Isis Gabriela Martínez López received PhD fellowship support from The Mexican Consejo Nacional de Ciencia y Tecnología (CONACyT). The funders had no role in study design, data collection and analysis, decision to publish, or preparation of the manuscript.

### Grant Disclosures

The following grant information was disclosed by the authors:
Mexican Consejo Nacional de Ciencia y Tecnología (CONACyT).

### Competing Interests

The authors declare that they have no competing interests.

### Author Contributions

- Isis Gabriela Martínez López conceived and designed the experiments, performed the experiments, analyzed the data, contributed reagents/materials/analysis tools, prepared figures and/or tables, authored or reviewed drafts of the paper, approved the final draft.
- Marloes van Den Akker conceived and designed the experiments, performed the experiments, analyzed the data, contributed reagents/materials/analysis tools, authored or reviewed drafts of the paper, approved the final draft.
- Liene Walk conceived and designed the experiments, performed the experiments, analyzed the data, contributed reagents/materials/analysis tools, authored or reviewed drafts of the paper, approved the final draft.
- Marieke M. van Katwijk contributed reagents/materials/analysis tools, authored or reviewed drafts of the paper, approved the final draft.
- Tjisse van Der Heide contributed reagents/materials/analysis tools, authored or reviewed drafts of the paper, approved the final draft.
- Brigitta I. van Tussenbroek conceived and designed the experiments, performed the experiments, contributed reagents/materials/analysis tools, prepared figures and/or tables, authored or reviewed drafts of the paper, approved the final draft.

## Field Study Permissions

The following information was supplied relating to field study approvals (i.e., approving body and any reference numbers):

Collection of plant tissue samples were approved by the Secretaria de Agricultura, Ganadería, Desarrollo Rural, Pesca y Alimentación, México (PPF/DGOPA-012/17).

## Data Availability

Data is available at the Open Science Framework: https://osf.io/2wfdp/?view_only=af37e8650bf54336bae3c91e95e754ae

## Supplemental Information

Supplemental information for this article can be found online at http://dx.doi.org/10.7717/peerj.7570#supplemental-information.

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
