# Peer review of "Nutrient availability induces community shifts in seagrass meadows grazed by turtles"

_PeerJ, doi:10.7717/peerj.7570_

## Round 0.1 · original submission · Major Revisions

In light of your Appeal, I have discussed the Decision I earlier made on this paper with the journal, and have agreed to revisit my earlier decision.

Below are my comments:

While the positive reviewer (Dr. Fourqurean) put his name on the paper, the other reviewer (anonymous) can also be considered a definite expert on the subject, and they raised several issues that need to be considered. The authors seem to primarily rely on the positive reviewer’s viewpoint and their own past research in their responses below.

The authors do need to at least address and consider the comments of reviewer 2, and not just say “they are wrong” or ignore comments because they are in opposition to reviewer 2.

For example:
Reviewer 2 wrote:
Among the complicating factors in the experimental design, the active turtle grazing that occurred in the simulated grazing plots while the experiment was in progress negates the use of results from this treatment and is only one example of uncontrolled treatments.
Our opinion: Absolute controlled treatments in the field are impossible. The fact that on 4 separate occasions in one patch (of the 18 times we applied clipping during the experiments), we found that the turtles had taken over our work of clipping could also be indicative that the manual manipulation that intents to copy the grazing behavior by turtles was sufficiently realistic. This has been a critic of one of earlier works of turtle grazing where we applied clipping (e.g. in Van Tussenbroek, B. I., & Morales, L. F. G.,2017, Grazing by green sea-turtles does not affect reproductive fitness in Thalassia testudinum. Aquatic Botany 141: 10-16)). Our visits to the experimental plots were very frequent, within the periods that seagrasses could fully grow back, and therefore we could always establish that the turtles had visited the sites, and their grazing and our clipping treatments were applied with almost the same frequency, thus creating minimal bias.
This information and justification should be added to the text.


Reviewer 2:
In addition, there was no justification provided for initiating simulated grazing in abandoned, naturally grazed patches (Long-term treatment). Because turtles don’t return to abandoned patches until they have recovered, this treatment and its results seem biologically irrelevant.
Our opinion: If the seagrass meadows are extensive and turtle density low, the turtles do not return after abandoning the patches. However, if the meadows are reduced, and the number of turtles is high, the turtles continue to visit and overgraze the plots, which has been reported in other studies including one by our team (Molina-Hernández A.L., Van Tussenbroek B.I., 2014, Patch dynamics and shifts in seagrass communities under moderate and high grazing pressure by green turtles. Marine Ecology Progress Series 517:143-157). Therefore, it is very relevant to understand what happens under a scenario of overgrazing. Dr. James Fourqurean found this a strong point of our study: “The paper describes mechanisms responsible for species shifts in plant communities under heavy grazing pressure. The work has special pertinence for seagrass meadows of the tropics, given recent reports of overgrazing and loss of these important ecosystems as sea turtle populations rebound.”
I could argue semantics, and discuss the “very pertinent” comment. Still, the point of reviewer 2 and the authors' response or thinking should be added to the text.


Reviewer 2:
It is also unclear why the authors clipped Syringodium and calcareous green algae, for which there was no evidence produced that these species are food resources for green turtles.
Our opinion: Although S. filiforme and calcareous green algae are not preferred food for the turtles, they do crop them occasionally and incidentally. To avoid bias, and to be certain of our outcomes, we found it better to experimentally crop all plants in the plots, knowing that we may mask potential shifts in species composition. In this context, Dr. James Fourquerean remarked that “Interestingly, these data show how grazing of all species without preference can still lead to changes in competitive hierarchy and therefore species composition of the grazed plant community compare to ungrazed plots.”
Yes, I can agree with this response, but the point of reviewer 1 remains to some degree, and this text needs to at the least be added to the paper.


Reviewer 2:
We also note that additional statistical analyses (the ANOVA table and post hoc results) are needed to further explain the densities and biomasses of each species. Post hoc tests carried out with their results shown on Figure 2 B would unlikely provide justification for the claim that grazing drives the shift seen in species composition, because the low p value for the ANOVA reported in the text has to be primarily due to the high value of the control and not differences among the grazing treatments.
Our opinion: Dr. James Fourqurean found the applied statistics solid. We considered his suggestion, and we calculated the relative contribution of early successional species to total biomass. These data also showed significant differences among the different levels of grazing treatments; which we will include when given the opportunity to resubmit our manuscript.
Who is “his” - reviewer 2 or Dr. Fourqurean?

We hope we have clarified that arguments of the second referee for a flawed design and interpretation are not valid in our opinion and are greatly in conflict with the comments of Dr. James Fourqurean. Therefore, we kindly request to change your decision and give us an opportunity to resubmit a revised version of the manuscript addressing the observations by the first referee, and the minor comments for the authors by the second referee, or to consider the opinion of a third reviewer.
My (editorial) response: Well, no, the flawed design comment is still valid. I can agree from experience that perfect experiments in the marine realm are tough, and am more than willing to give the authors the benefit of the doubt on this subject. Still, I believe reviewer 2 has some valid points given their level of expertise on the system you are studying.

Thus, I would like to ask the authors to perform a major revision. I aim to send the revised version back to both reviewers 1 and 2; thus, they really need to consider adding explanations into some of their methodologies as they have kindly done in the letter above. If the reviewers are still opposed, I will bring a third reviewer as kindly suggested by the authors.

Please let me know if you have any questions or concerns.

· Appeal

Appeal

The reason of this letter is a request to re-examine your decision, because both reviewers found the topic of interest for the readers of PeerJ, and the comments of fist reviewer, Dr. James Fourqurean, a highly recognized expert in ecology and biogeochemistry of seagrasses, were positive and mainly concerned details but not in-depth objections. Strangely enough, the reasons for rejection of our work by the second reviewer were often in conflict with the comments of Dr. James Fourqurean. Below we would like to address the objections raised by the second reviewer:

Reviewer 2:

Among the complicating factors in the experimental design, the active turtle grazing that occurred in the simulated grazing plots while the experiment was in progress negates the use of results from this treatment and is only one example of uncontrolled treatments.

Our opinion: Absolute controlled treatments in the field are impossible. The fact that on 4 separate occasions in one patch (of the 18 times we applied clipping during the experiments), we found that the turtles had taken over our work of clipping could also be indicative that the manual manipulation that intents to copy the grazing behavior by turtles was sufficiently realistic. This has been a critic of one of earlier works of turtle grazing where we applied clipping (e.g. in Van Tussenbroek, B. I., & Morales, L. F. G.,2017, Grazing by green sea-turtles does not affect reproductive fitness in Thalassia testudinum. Aquatic Botany 141: 10-16)). Our visits to the experimental plots were very frequent, within the periods that seagrasses could fully grow back, and therefore we could always establish that the turtles had visited the sites, and their grazing and our clipping treatments were applied with almost the same frequency, thus creating minimal bias.

Reviewer 2:

In addition, there was no justification provided for initiating simulated grazing in abandoned, naturally grazed patches (Long-term treatment). Because turtles don’t return to abandoned patches until they have recovered, this treatment and its results seem biologically irrelevant.

Our opinion: If the seagrass meadows are extensive and turtle density low, the turtles do not return after abandoning the patches. However, if the meadows are reduced, and the number of turtles is high, the turtles continue to visit and overgraze the plots, which has been reported in other studies including one by our team (Molina-Hernández A.L., Van Tussenbroek B.I., 2014, Patch dynamics and shifts in seagrass communities under moderate and high grazing pressure by green turtles. Marine Ecology Progress Series 517:143-157). Therefore, it is very relevant to understand what happens under a scenario of overgrazing. Dr. James Fourqurean found this a strong point of our study: “The paper describes mechanisms responsible for species shifts in plant communities under heavy grazing pressure. The work has special pertinence for seagrass meadows of the tropics, given recent reports of overgrazing and loss of these important ecosystems as sea turtle populations rebound.”

Reviewer 2:

It is also unclear why the authors clipped Syringodium and calcareous green algae, for which there was no evidence produced that these species are food resources for green turtles.

Our opinion: Although S. filiforme and calcareous green algae are not preferred food for the turtles, they do crop them occasionally and incidentally. To avoid bias, and to be certain of our outcomes, we found it better to experimentally crop all plants in the plots, knowing that we may mask potential shifts in species composition. In this context, Dr. James Fourquerean remarked that “Interestingly, these data show how grazing of all species without preference can still lead to changes in competitive hierarchy and therefore species composition of the grazed plant community compare to ungrazed plots.”

Reviewer 2:

We also note that additional statistical analyses (the ANOVA table and post hoc results) are needed to further explain the densities and biomasses of each species. Post hoc tests carried out with their results shown on Figure 2 B would unlikely provide justification for the claim that grazing drives the shift seen in species composition, because the low p value for the ANOVA reported in the text has to be primarily due to the high value of the control and not differences among the grazing treatments.

Our opinion: Dr. James Fourqurean found the applied statistics solid. We considered his suggestion, and we calculated the relative contribution of early successional species to total biomass. These data also showed significant differences among the different levels of grazing treatments; which we will include when given the opportunity to resubmit our manuscript.

We hope we have clarified that arguments of the second referee for a flawed design and interpretation are not valid in our opinion and are greatly in conflict with the comments of Dr. James Fourqurean. Therefore, we kindly request to change your decision and give us an opportunity to resubmit a revised version of the manuscript addressing the observations by the first referee, and the minor comments for the authors by the second referee, or to consider the opinion of a third reviewer.

Thank you for receiving this manuscript, and we hope that you will consider it for review. We appreciate your time and look forward to your response.S


· · Academic Editor

Reject

I have now heard back from two expert reviewers. While both are positive about the amount of work you have put into the study, Reviewer 2 has noted several points about your research that, in my opinion, cannot be addressed in the current form of your work, and therefore I am rejecting this submission. I would welcome a paper that addresses all of Reviewer 2's concerns as a new submission, and hope that despite the rejection, the comments provided here can help you make your work more robust. Thank you for submitting to PeerJ.

·

Basic reporting

This paper is very well-written in clear, unambiguous professional English. The introduction does a great job motivating the study, with appropriate literature from both marine and terrestrial literature. The paper generally follows the PeerJ standards as outline in the instructions for authors, with the exception that the manuscript has no Conclusions section. This does not trouble me as a reviewer and reader, but it may require reformatting to end with a conclusions section if directed by the editors. The figures as they appear in the paper are high-quality and relevant, but in the comments I have a suggestion for ether modifying or replacing Fig 2. And, although the authors did provide the near-raw data in a link in their submission materials, there is no link to the raw data nor instructions how to find them in the manuscript. Also, the data are lacking the metadata that will allow future readers to decipher the meaning of the data columns.

Experimental design

This research is within the scope of PeerJ. The paper describes mechanisms responsible for species shifts in plant communities under heavy grazing pressure. The work has special pertinence for seagrass meadows of the tropics, given recent reports of overgrazing and loss of these important ecosystems as sea turtle populations rebound. This paper is the first to point to mechanisms leading to change in species dominance under grazing in seagrasses. The investigators were rigorous and designed experiments well to test their hypotheses. As far as I can tell from the manuscript, the authors have a high ethical standard. I do have to take their word for it that they obtained all necessary permits from the Mexican authorities, but I have no reason to doubt them nor do I have enough expertise with local laws to know if they got all of the permissions they were required to have. The methods are well-described.

Validity of the findings

I found the work to be statistically sound and based on theoretical predictions. This paper links the seagrass grazing literature to more basic literature on the competitive interactions among plants to shed light on the mechanisms for changing species dominance under grazing. Interestingly, these data show how grazing of all species without preference can still lead to changes in competitive hierarchy and therefore species composition of the grazed plant community compare to ungrazed plots.

Additional comments

I like the paper very much. Below I list my specific comments and questions that arose as I was reading the paper.
Line 75 – Rotational grazing was proposed long before the Molina-Hernandez and van Tussenbroek 2014 paper, by, among others, Bjorndal, Susan Williams, Gordon Thayer and Jay Zieman.
Line 98 – be more clear about the nature of light available to seagrasses. In clear waters, light availability at the top of the canopy can be high – but the canopy itself can produce light limitation in dense meadows. This would put the statement about light attenuation of the canopies on line 100 into context.
Line 141. Please revisit the terminology used for the different grazing treatments. Both what are now described as short- and medium-term grazing treatments were done over the same time interval, so there is no difference in the term of them. Rather, the grazing interval was different among the these treatment. Perhaps call them short term frequent and short-term, infrequent grazing treatments?
Line 148. Shears d not really mimic the size of a turtle bite; rather they leave behind ungrazed shoot bases that mimic those following turtle grazing.
Line 182-end of paragraph. Please state the limits of detection of the chemical analysis methods. And, did the authors check for sulfide interference in these colorimetric analyses? My experience has been that sulfides need to be purged before porewater analyses because they poison colorimetric determinations of ammonium and phosphate.
Line 190. Technically, quadrats have to be squares (hence the name QUADrat). Another name for “circular quadrat”?
Line 263 and throughout. Do the alanlyses really justify reporting sulfide (and other parameters) to 2 decimal places? And, at 380 uM, sulfides would likely cause interferences in the phosphate ad ammonium determinations.
Line 290 and throughout. I’d rather see “N concentrations” rather than “N%” or the redundant “N% concentrations” since % is just the unit of expressing N concentration.
Line 305. “species composition” rather than “specific composition” since the adjective “specific” could mean the composition at a particular time or place.
Line 322. C:N ratio, even at the latitude of this study, is seasonal, with nutrient limitation signals generally only evident during times of peak growth rate. Did the authors consider seasonality of the seagrass stoichiometry when designing the study?
Line 329. I believe that almost all studies of the change in N content of seagrasses in response to grazing find no decrease in N content driven by grazing (examples, the Moran and Bjorndal papers, Fourqure et al 2010, others). The speculation in the Thayer et al 1984 paper that depleted N in tissues should serve as a patch abandonment cue was made with no supporting data, but this somehow became part of the paradigm, despite decades of experimental work that shows N content does not decrease in the face of grazing, simulated or natural.
Line 366. Awkward to start a sentence with “But”, I suggest linking this sentence to the previous one with ‘…), however, …”
Line 381, and figure 2. One logical problem in the data is that ungrazed plots have higher biomass of the early successional species that should be out-competed by the slow growing climax species than in clipped plots. How can this be? Perhaps it would be good to calculate the relative contribution of early sucessional species to total biomass ((Halodule+Syringodium+Calcareous Algae)/ (Halodule+Syringodium+Calcareous Algae+Thalassia). Shouldn’t this relative contribution to total biomass be the real critical response to changing nutrient competition?
Literature cited - did not check all references, but I note that there was no Heithaus et al 2014 in the literature cited section. Please check to make sure all cited papers are in the literature cited section.

Reviewer 2 ·

Basic reporting

In the manuscript Nutrient availability induces community shifts in seagrass meadows grazed by turtles López et al seek to add to the growing literature on the effects that increasing and range-expanding populations of herbivorous green sea turtles can have on seagrass meadows. In this manuscript Lopez et al report that increased grazing pressure induced by both simulated and natural green turtle grazing led to increased pore-water ammonium concentrations and shifts in seagrass species composition from a long-lived (turtlegrass Thalassia testudinum) to early successional species (manatee grass Syringodium filiforme & rhizophytic algae). They also found no differences in organic sediment content among grazing treatments.
While there is a need for additional information on the effects on sediments and pore water nutrient content affected by green turtle grazing, the flaws in this study’s experimental design prevent sound conclusions from being drawn. Below we consider only a few of our concerns, but they are sufficient to support our assessment that the paper is fatally flawed.

Experimental design

Among the complicating factors in the experimental design, the active turtle grazing that occurred in the simulated grazing plots while the experiment was in progress negates the use of results from this treatment and is only one example of uncontrolled treatments. In addition, there was no justification provided for initiating simulated grazing in abandoned, naturally grazed patches (Long-term treatment). Because turtles don’t return to abandoned patches until they have recovered, this treatment and its results seem biologically irrelevant. It is also unclear why the authors clipped Syringodium and calcareous green algae, for which there was no evidence produced that these species are food resources for green turtles.
We also note that additional statistical analyses (the ANOVA table and post hoc results) are needed to further explain the densities and biomasses of each species. Post hoc tests carried out with their results shown on Figure 2 B would unlikely provide justification for the claim that grazing drives the shift seen in species composition, because the low p value for the ANOVA reported in the text has to be primarily due to the high value of the control and not differences among the grazing treatments.

Validity of the findings

Among the complicating factors in the experimental design, the active turtle grazing that occurred in the simulated grazing plots while the experiment was in progress negates the use of results from this treatment and is only one example of uncontrolled treatments. In addition, there was no justification provided for initiating simulated grazing in abandoned, naturally grazed patches (Long-term treatment). Because turtles don’t return to abandoned patches until they have recovered, this treatment and its results seem biologically irrelevant. It is also unclear why the authors clipped Syringodium and calcareous green algae, for which there was no evidence produced that these species are food resources for green turtles.
We also note that additional statistical analyses (the ANOVA table and post hoc results) are needed to further explain the densities and biomasses of each species. Post hoc tests carried out with their results shown on Figure 2 B would unlikely provide justification for the claim that grazing drives the shift seen in species composition, because the low p value for the ANOVA reported in the text has to be primarily due to the high value of the control and not differences among the grazing treatments.

Additional comments

Specific comments:
Introduction
● Line 49: Lal et al. 2010 also reports changes in seagrass bed composition due to turtle grazing
● Line 102: “Sensu Williams (1990)...” into “Williams (1990) reported...”
Materials and Methods:
● Combine Simulated grazing and “Seagrasses and rhizophytic algae” sections
● Line 133: i.e.instead of “species like”
● Lines 141-142: unnecessary capital letters in treatment names
● Lines 142-143: Grazing an abandoned patch is not a realistic treatment
● Line 146-147:there is no evidence turtles graze on Syringodium is a primary food source, most likely just occasional, incidental grazing (Molina-Hernández & van Tussenbroek (2014)), clipping it does not mimic natural grazing
● Line 154-156: time needed for seagrasses to grow enough would depend on the time of year. References show no evidence of Thalassia needing 15 days to regrow >5 cm in any of the references except Moran & Bjorndal 2007 says “Clipping interval ranged from 12 to 38 days because growth rates varied with temperature”
● Line 156: replace “during” with “for”
● Line 161: “...the leaves in these patches did not have bite 161 marks of the turtles which are clearly visible” reconsider sentence structure
● Line 168: “...in February 2016.” say what this means timeline-wise
● Line 175-176: again, tell us what this means for your timeline. Halfway into the experiment? 3 weeks in?
● Line 190-191: replace symbols for meaning
● Line 233: lowercase o in “One-way ANOVA”
● Line 240-241: leverage is not used correctly. Stop sentence after “...were removed” The term “outliers” is enough justification.
● Line 244: same as line 210-241

Results
● Combine Simulated grazing and “Seagrasses and rhizophytic algae response to grazing pressure” sections
● Lines 251-252: natural grazing co-occurring with simulating grazing would negate the results as the grazing regime is not followed
● Line 247: if there are no results to report in the simulated turtle grazing results section, don’t include the section
● Lines 273-277: There are no significant differences, and the differences are not enough to assert this trend. Include ANOVA table and post hoc results because low p value reported in the text can be driven mainly due to the control
● Line 278: Fig. 2
● Line 283: two-way
Discussion:
● Line 307: write out the full scientific name and common name every time you start a new section
● Line 382-386: irrelevant because turtles only incidentally graze Syringodium
● Line 388-390: Lal et al. 2010 reported the same for turtle grazing
Tables:
● 1
○ “ ...undisturbed until the end...”
○ “Experimental grazing (clipping maintenance)” as the column header is unclear, suggest Experimental grazing duration
○ If “Does not occur naturally in patch rotational grazing because turtles do not return to abandoned patches” is true of the long-term treatment in previously grazed grass, what is the justification for this treatment?
● 2: Include what the error value is
● 4: nitrogen (N) and carbon (C)
Figures:
● 1: include an explanation of the dotted line
● 2: extra parenthesis
● 3: include legend on the graph
● 4: graphic is too complex and not understandable

Overall recommendation: Clearly, a great deal of work was done to gather the data reported in this manuscript. However, the lack of proper control of treatments, along with other issues noted above, render the results and conclusions in this paper unreliable, and a recommendation that the paper be rejected the only one possible. There may be a way to use portions of the data to produce a more qualitative, narrative type manuscript but this will require a significant amount of work.

---

## Round 0.2 · accepted · Accept

I have gone over your paper in detail, along with your responses. In particular, the added explanations in the Materials and Methods serve the paper well.

There are some small English edits here and there, and I have included an annotated MSWord file that the editing office will send you. Please ensure edits are done at the Proof stage if not earlier.

Thus, in summary, I am happy to now accept your work, and thank you for your hard work in revising this manuscript.